# Exposure of the Gestating Mother to Sympathetic Stress Modifies the Cardiovascular Function of the Progeny in Male Rats

**DOI:** 10.3390/ijerph20054285

**Published:** 2023-02-28

**Authors:** Beatriz Piquer, Diandra Olmos, Andrea Flores, Rafael Barra, Gabriela Bahamondes, Guillermo Diaz-Araya, Hernan E. Lara

**Affiliations:** 1Centre for Neurobiochemical Studies in Neuroendocrine Diseases, Laboratory of Neurobiochemistry, Department of Biochemistry and Molecular Biology, Universidad de Chile, Santiago 8380492, Chile; 2Centro de Investigación Biomédica y Aplicada (CIBAP), Escuela de Medicina, Facultad de Ciencias Médicas, Universidad de Santiago de Chile, Santiago 9170020, Chile; 3Department of Chemical Pharmacology and Toxicology, Faculty of Chemistry and Pharmaceutical Sciences, Universidad de Chile, Santiago 8380492, Chile

**Keywords:** gestational programming, cardiovascular, norepinephrine, beta receptors

## Abstract

Background: Sympathetic stress stimulates norepinephrine (NE) release from sympathetic nerves. During pregnancy, it modifies the fetal environment, increases NE to the fetus through the placental NE transporter, and affects adult physiological functions. Gestating rats were exposed to stress, and then the heart function and sensitivity to in vivo adrenergic stimulation were studied in male progeny. Methods: Pregnant Sprague–Dawley rats were exposed to cold stress (4 °C/3 h/day); rats’ male progeny were euthanized at 20 and 60 days old, and their hearts were used to determine the β-adrenergic receptor (βAR) (radioligand binding) and NE concentration. The in vivo arterial pressure response to isoproterenol (ISO, 1 mg/kg weight/day/10 days) was monitored in real time (microchip in the descending aorta). Results: Stressed male progeny presented no differences in ventricular weight, the cardiac NE was lower, and high corticosterone plasma levels were recorded at 20 and 60 days old. The relative abundance of β1 adrenergic receptors decreased by 36% and 45%, respectively (*p* < 0.01), determined by Western blot analysis without changes in β2 adrenergic receptors. A decrease in the ratio between β1/β2 receptors was found. Displacement of ^3^H-dihydroalprenolol (DHA) from a membrane fraction with propranolol (β antagonist), atenolol (β1 antagonist), or zinterol (β2 agonist) shows decreased affinity but no changes in the β-adrenergic receptor number. In vivo exposure to ISO to induce a β-adrenergic overload provoked death in 50% of stressed males by day 3 of ISO treatment. Conclusion: These data suggest permanent changes to the heart’s adrenergic response after rat progeny were stressed in the uterus.

## 1. Introduction

During gestation, all organisms adapt to the local conditions existing in the uterus. In the placental environment, they are exposed to different stimuli that could be dangerous if they are chronic. If there are no changes in their genetic load, this adaptation to the surrounding environment will produce changes in a single generation, which is much faster than genetic adaptation [1,2,3]. Increasing evidence supports that these changes in the offspring could involve changes that could have transgenerational effects. In this regard, if chronic stress occurs during the gestation period, this stress modifies the placental environment to produce changes in the embryo, affecting the development of organs, the physiology, and the metabolism of the organism (fetal programming) [4,5]. Various studies in animals have shown the effects of fetal programming. In Wistar rats, maternal protein restriction during gestation and lactation results in increased levels of cardiac fibrosis in the adult offspring [6], lower concentrations of cardiac β1-adrenergic receptors, and a reduced baseline heart rate [7]. Similar effects have been found in young animals that were stressed during the gestation period. For example, in Sprague–Dawley rats, adult pups, which had suffered gestational stress, had a more significant increase in blood pressure and heart rate due to stress [8]. Sympathetic stress stimulates norepinephrine (NE) release from sympathetic nerves. During pregnancy, it modifies the fetal environment, increasing the access of NE to the fetus through the placental NE transporter and probably affecting adult physiological functions [9]. The effects of chronic stress on people’s health and, in particular, on embryos are just beginning to be analyzed, making it important to understand the effects of fetal programming. Regarding the stress response, the activation of the SNS is observed in “fight or flight” situations, as would be in the case for a cold situation without hypothermia [1,10,11]. Along with the increased stress levels in the population, the frequency and mortality associated with cardiovascular diseases in humans have increased, with a direct association between stress and these diseases. Currently, cardiovascular diseases are considered one of the leading causes of death in the adult population, and their prevalence is expected to continue to increase [12].

Our laboratory works with an intermittent chronic cold stress protocol (4 °C/3 h/day) applied to the gestating rat throughout the gestation period. This stress model produces a selective sympathetic response and a decrease in the placental norepinephrine transporter (NET), which produced an overexposure to NE of the fetus during the gestation period [10,11,13]. This stress modifies the reproductive function of the adult female progeny with the induction of a polycystic ovarian phenotype (presence of ovarian cysts, hyperandrogenic condition [14], insulin resistance in the ovary [15], and modified reproduction of the progeny [16,17]. Interestingly, cold stress could have broad implications for the development of the offspring. Prenatal cold stress has been associated with an altered pattern of phosphorylation in ERK1/2 and induction of BDNF expression in the hippocampus [18]. Additionally, prenatal cold stress increases the blood pressure in the offspring due to lower levels of Dopamine D1 vascular receptor [19]. Based on these considerations regarding the effects of gestational stress in the female progeny and the possible impact of a high noradrenergic state during gestation and cardiovascular disease in the progeny, we hypothesized that the chronic gestational exposure to higher levels of NE, due to cold stress, could also program cardiovascular alterations in the male adult offspring. During this experimental protocol, both males and females were born; the results of some of the females studied correspond to already published reproductive studies [17]. In the present report, we present the effects on the cardiovascular system of the male progeny.

## 2. Animals and Experimental Design

Primiparous female Sprague–Dawley rats weighing between 250 and 300 g, kept at 20 °C and with a 12:12 light–dark cycle, were used. Food and water were constantly and freely available. Mating procedure: Rats were checked for estrous cyclic activity to determine the proestrus day. On the night of the proestrus phase (when ovulation occurs), the rats were mated with males of proven fertility. The following morning, the rats were checked for the presence of a vaginal sperm plug. Rats with a vaginal plug were assigned as day 0 of pregnancy. A total of 20 pregnant rats were randomized into two groups of 8 control rats and 12 stressed rats. All the rats of the male progeny were euthanized at the age shown in Figure 1, depending on the experimental protocol used. The rats were euthanized by decapitation at the end of the experiments; the hypothalamus (medial basal hypothalamus), heart (ventricular tissue), and plasma of each rat were collected. Tissues were weighed on a laboratory scale with a sensitivity of up to 1 mg. Decapitation was performed according to the AVMA Guidelines for the Euthanasia of Animals (2020 Edition) [20]. This was performed by specialized personnel and approved by the Bioethics Committee of the Faculty of Chemistry and Pharmaceutical Sciences at the University of Chile (protocol number: CBE2016-13 to BP and CBE2017-05 to HL). All procedures presented in this paper complied with the National Institutes of Health guide for the care and use of Laboratory animals (NIH Publications No. 8023, revised 1978) and national guidelines (CONICYT Guide for the Care and Use of Laboratory Animals). Figure 1 graphically shows the different experimental protocols used in the biological assays and the total number of rats.

### 2.1. Gestational Stress Induction

From day 1 of pregnancy, pregnant female rats from the experimental group were exposed daily at 4 °C for 3 h a day throughout the gestation period. The control group was kept at a constant temperature of 20 °C throughout the gestation period. Once rats were born, their sex was determined, and they were assigned to foster mothers up to 20 days old. The progeny were regulated to 12 rats each, with half of the pups being male and the other being female (to maintain the average of pups naturally born and ensure sufficient nursery. The female progeny was included in another protocol to study fertility during a multigeneration study [17], and the male progeny of the present report was studied at 20 and 60 days old. The chronic cold stress protocol for pregnant rats was the same as we previously used for adult rats [9,16]. 

### 2.2. Adrenergic Overload Protocol

When male progeny reached 50 days old, we used two groups with 10 control and 10 rats exposed to gestational stress during fetal growth. Both the control and stressed rats received a daily injection of isoproterenol (ISO, β-adrenergic agonist, Sigma Chemical Co., St. Louis, MO, USA) at a daily dose of 1 mg/kg of weight for 10 days. This dose produced heart hypertrophy after 10 days in control untreated rats [21]. ISO was dissolved in saline supplemented with ascorbic acid (Sigma Chemical Co., St. Louis, MO, USA) at a concentration of 0.5 mM to prevent the oxidation of ISO. Each day, solution was freshly prepared, and the solution was sterilized by passage through a 0.22 µm filter. We also studied two groups (control and stressed) injected with saline as controls of the ISO group). These animals were euthanized at 60 days old after the 10th dose of ISO, before they resisted adrenergic overload with ISO exposure. For the in vivo recording of arterial pressure, we implanted a microtransducer (TRM54P, from Millar, Inc., Houston, TX, USA) coupled to a PowerLab DAQ (ADInstruments, Sydney, Australia). Fifty-day-old stressed and control male rats were anesthetized with an i.m. dose of ketamine 60 mg/kg/xylazine 10 mg/kg solution under aseptic conditions. A transverse mid-lumbar incision of ~1.5 cm long was made on anesthetized rats, and the microdevice was implanted in the peritoneal cavity. A cannula was connected to the abdominal aorta and sealed with instant glue. Rats were maintained in postoperative observation for 3 days before starting to record the pressure. The device was wirelessly charged through a platform that also received diastolic and systolic pressure signals. The signals were recorded through the program incorporated into the system.

### 2.3. Quantification of NE Levels

The quantification of NE concentration in the heart, hypothalamus, and plasma was performed by homogenizing all the tissues in 10 volumes of 0.2 M perchloric acid in ice. The NE levels were quantified using the Norepinephrine ELISA kit—Research^®^. The minimal detectable value was 1.3 pg/mL, with a range of 0.2–32 ng/mL (IMMUSMOL, Pessac, France). NE was extracted using a cis-diol-specific affinity gel, acylated, and then derivatized enzymatically. The antigen was bound to the solid phase of the microtiter plate. The derivatized standards, controls, and samples and the solid-phase-bound analyte competed for a fixed number of antiserum binding sites. The antibody bound to the solid phase was detected using an anti-rabbit IgG-peroxidase conjugate and 3,3′,5,5′-tetramethylbenzidine (TMB) as a substrate. The reaction is monitored at 450 nm. The results are expressed as the total amount of NE in ng per ovary. The sensitivity was 2 pg/mL, and the intra- and interassay variability were 8.4 and 8.0%, respectively. The cross-reactivity found was 0.14% for adrenaline and 1.8% for dopamine.

### 2.4. Quantification of Serotonin in the Medial Basal Hypothalamus (MBH) by ELISA

Before determining serotonin, the MBH was weighed and homogenized by ultrasound using 1 mL of dilution buffer, always on ice. The homogenate was centrifuged at 13,500 rpm at 4 °C for 15 min, and the supernatant was recovered and diluted 20 times with dilution buffer for quantification with the ELISA BA E-5900 kit (IMMUSMOL, Pessac, France), which has a sensitivity of 5 pg/mL. No significant cross-reactivity was observed with serotonin analogs such as tryptamine and melatonin. In addition, according to the manufacturer, there is no cross-reactivity with 5-hydroxyindole acetic acid, phenylalanine, histidine, tyramine, and 5-hydroxytryptophan.

### 2.5. Quantification of Plasma Levels of Corticosterone by ELISA

Trunk blood obtained at the moment of euthanasia was centrifuged, and the plasma was used for the assay according to the manufacturer’s instructions. The corticosterone levels were determined using an enzyme immunoassay (Alpco Diagnostic, Windham, NH, USA). We found an intra-assay variation of <8.3% and an inter-assay variation of <12.4%. The minimal detectable value was 4.1 ng/mL.

### 2.6. Binding of [^3^H]dihydroalprenolol to Cardiac Membranes

#### 2.6.1. Preparation of the Crude Membrane Fraction

One hundred milligrams of the basal area of the heart tissue were weighed and homogenized in an ultraturrax with 0.7 mL of 20 mM Tris-HCl/0.25 M Sucrose buffer at pH 7.4. Then, it was centrifuged at 13,000 rpm for 20 min at 4 °C. The supernatant was discarded, and the precipitate was suspended in 0.7 mL of 20 mM Tris-HCl/0.25 M sucrose buffer at pH 7.4 and centrifuged again at 13,000 rpm for 20 min at 4 °C. The supernatant was removed, the precipitate was suspended in 0.7 mL of 20 mM Tris-HCl/10 mM MgCl_2_ buffer pH 7.4 without sucrose, and the protein concentration was measured by the Bradford method [22].

#### 2.6.2. Assay for β-Adrenergic Receptors

Glass tubes were used, working in duplicate, which contained the membrane samples obtained after tissue homogenization (volume equivalent to 300 μg of protein) and 20 μL of 10 nM ^3^H-DHA in a total volume of 0.2 mL. The assay contained 0.5–40 nM 3Hdihydroalprenolol (92.1 Ci/mmol, NEN DuPont). Nonspecific binding was assessed: 20 μL of 10^−3^ M Propranolol (only for the tubes corresponding to nonspecific binding) and 20 mM Tris-HCl/10 mM MgCl_2_ buffer pH 7.4 without sucrose (in a sufficient quantity to complete 200 μL as the final volume). The tubes were incubated for 30 min at 37 °C. Then, they were filtered with Whatman GF/C filters in a vacuum system and washed three times with 2 mL of 20 mM Tris-HCl/10 mM MgCl_2_ buffer pH 7.4 without sucrose with 10–55 M propranolol. Finally, the filters were placed in scintillation vials, 0.5 mL of distilled water and 4 mL of scintillation mix were added to each vial, each tube was vortexed, and the radioactivity was measured in a scintillation counter 24 h later [23,24]. Statistical analyses were performed using GraphPad Prism 6 (GraphPad Software, San Diego, CA, USA).

### 2.7. Determination of Cardiac β_1_- and β_2_-Adrenergic Receptors by Western Blot

Protein extraction: Approximately 100 mg of cardiac tissue was weighed and homogenized in glass using RIPA buffer (Tris-HCl pH 7.2 10 mM, Triton X-100 1%, NaCl 150 mM, SDS 0.1%), a process carried out on ice. The obtained product was centrifuged at 13,000 rpm for 20 min at 4 °C. The pellet was discarded, the supernatant was recovered, and the protein concentration was subsequently measured using the Bradford method [22].

Electrophoresis in polyacrylamide gels: Gels were loaded with 20 µg of protein. Electrophoresis was performed at a constant voltage of 120 V in 1× electrophoresis buffer (Tris-Base 0.3% (*w/v*), glycine 1.44% (*w/v*), SDS 0.1% (*w/v*); pH 8, 6) for 90 min. The proteins were transferred to a pore nitrocellulose membrane of 0.45 μm at 400 mA constant for 90 min in transfer buffer (Tris-Base 0.5% (*w/v*), glycine 2.4% (*w/v*), and methanol 20% (*v/v*)). After the transfer was completed, the membrane was stained with 5% Ponceau Red to verify the correct transfer. Subsequently, the staining was eliminated with washings with tap water and three washes with TBST 0.1% (Tris-Base 0.224% (*p/v*), NaCl 0.8% (*p/v*), Tween 0.1% (*v/v*), adjust to pH 7.6), each for 5 min. Nitrocellulose membranes were incubated with gentle shaking with 5% milk in 0.1% TBST at room temperature to block nonspecific binding sites. We performed three washes with 0.1% TBST, each for 5 min, and subsequently, incubation was administered overnight at 4 °C with primary antibodies to the β1- or β2-adrenergic receptor using a dilution of 1:3000 and 1:2500, respectively. The antibodies used for beta-1 were a polyclonal anti-beta 1 adrenergic receptor antibody (ab3442, from ABCAM) and a recombinant anti-beta 2 adrenergic receptor antibody (ab182136, from ABCAM). Both antibodies were diluted in TBST 0.1 and 5% milk to avoid nonspecific binding. After incubation, the membrane was washed three times for 5 min with TBST 0.1% and incubated for 1 h, with shaking and at room temperature and with the secondary antibody (anti-rabbit polyclonal antibody from ABCAM), in both conditions using a dilution of 1:10,000 in TBST 0, 1%. The membrane was washed three times for 5 min each time. GAPDH, a highly conserved protein present in similar amounts, independent of the sample type, was used as a loading control. The primary antibody was incubated for 1 h using a dilution of 1:40,000 and subsequently incubated with a secondary anti-rabbit antibody incubated at 1:10,000 for 45 min. Using the GeneSys G-Box system (Syngene Headquarters, Frederick, MD, USA), chemiluminescence was developed using the EZ-ECL reagent. The intensity of the bands was evaluated using the ImageJ program.

### 2.8. Morphometric Analysis

The medial portion of the left ventricle was fixed in Bouin, dehydrated at 4 °C for 24 h, and washed with 70% ethanol, and the tissue was embedded in paraffin. We used 5 µm slices stained with hematoxylin-eosin and tricromo Masson and examined them under a light microscope. We measured the area and perimeter of the cardiomyocytes according to Nakamura et al. [25]. Briefly, we analyzed four contiguous slices, and when an apparent nucleus appeared, the area and perimeter were measured with ImageJ software (Version 1.53t). All analyses were performed with at least 100 cells per heart.

### 2.9. Statistical Analysis

The data are expressed as the mean ± SEM. Statistical analyses were performed using GraphPad Prism 6 (GraphPad Software, San Diego, CA, USA). To test for significant differences between two groups, we used Student’s *t*-test and the Mann–Whitney test. To test for significant differences among more than two groups, we used one-way ANOVA, followed by the Newman–Keuls post hoc test to check for the differences between pairs of data. The number of animals for all experiments was calculated as the minimum number of animals according to the variability of the experimental procedures. The minimum number of animals was calculated according to Equation [26]:$$n=\frac{2{\left(Z\alpha +Z\beta \right)}^{2}\times S^{2}}{d}$$
where n is the number of animals for each condition, *S* is the standard deviation, *d* is the difference needed to obtain statistical significance, *Zα* is the probability of type I error (significance), and *Zβ* is the probability of type II error (power). We proposed *α* = 0.05, i.e., the probability of finding a statistically significant difference was 0.05; *β* = 0.3, the probability of having a difference between the populations; the intrapopulation variation was 0.2; and *d*, the smallest difference in the population, was 0.11. Thus, we obtained n = 4.5. Therefore, to obtain a statistically significant difference of *p* < 0.05, we needed to use four or five animals per study group.

## 3. Results

### 3.1. Animal and Heart Relative Weight

Stress did not modify the gestation days, pup numbers, or male and female pup proportions (Table 1). Rats’ weight and hearts were also presented at 20 and 60 days of age. No changes in rats’ weight either at 20 days old or at 60 days old were found. The heart weight was normalized to animal weight, but no differences were found.

### 3.2. Hypothalamic, Plasma, and Cardiac Neurotransmitters and Plasma Corticosterone

Figure 2 shows that gestational stress increased the concentration of NE in the medial basal hypothalamus of the progeny at 20 and 60 days of age. No changes due to age were found in control and stressed rats. Hypothalamic serotonin was also increased at 20 days old, but no changes were found at 60 days old due to gestational stress. Both control and stressed rats had increased serotonin concentrations due to age (Figure 2B). No changes in plasma NE were found between control and stressed rats (Figure 2C), although a decrease in NE in the hearts of 20- and 60-day-old stressed rats was found (Figure 2D). The plasma corticosterone levels were also increased in stressed rats and maintained at 20 and 60 days old (Figure 2E).

### 3.3. Changes in β1 and β2 Adrenergic Receptor Protein Abundance 

β-adrenergic receptor levels were studied by Western blot in heart tissue. The results showed that the β1-adrenergic receptor abundance decreased in the hearts of stressed rats at both 20 and 60 days old (Figure 3A), whereas no changes were observed in β2 adrenergic receptor abundance in stressed versus control rats (Figure 3B). Consequently, the cardiac level ratio of β1/β2 receptors was decreased in stressed rats (Figure 3C). 

### 3.4. Concentration of β-Adrenergic Binding Sites

The β-adrenergic receptor affinity (IC50) and receptor numbers were measured in the heart tissue of the control and stressed male rats sacrificed at 60 days old. Competition curves were performed to see the possible effects of the gestational stress protocol on the affinity of these receptors using the radioligand technique in fractions of cardiac membranes from male rats sacrificed at 60 days old. Increasing concentrations (10^−3^–10^−8^ M) of propranolol (a nonselective β-adrenergic antagonist, which allows us to analyze the affinity of the total β-adrenergic receptor), atenolol (β1-adrenergic receptor selective antagonist), and zinterol (β_2_-adrenergic receptor selective antagonist) (Figure 4A–C), and the IC50 (necessary dose of a drug to obtain 50% displacement) were used for each case. A significant increase was observed in the IC50 of propranolol, atenolol, and zinterol (Figure 4D); however, no changes in the β-adrenergic receptor number were observed (Figure 4E).

### 3.5. Effects of Isoproterenol on Ventricle Size and Cardiomyocyte Area in Gestationally Stressed Rats

We analyzed the ventricular size and cardiomyocyte area to analyze whether exposure to gestational stress also modified the morphological parameters of heart ventricles and cardiomyocyte size. As shown in Figure 5A, control and stressed rats did not present changes in ventricle size at 60 days old; however, after ISO treatment for 10 days, an increased ventricle weight was observed, which was higher in stressed rats. In addition, the cardiomyocyte area was similar in control and stressed rats (Figure 5B,C), although, after ISO challenge for 10 days, we found that control and stressed rats treated with ISO had an increased cardiomyocyte area, which was significantly higher in stressed rats.

### 3.6. Cardiac Functional Parameters in Stressed Offspring Challenged with Isoproterenol Overload

Since the main differences were found in the affinities of β-adrenergic receptors and the ventricle size and cardiomyocyte area of stressed animals, we performed a functional analysis to verify the pathophysiological impact of exposure to an adrenergic loading protocol. In Figure 6, we show that 50% of the progeny of rats exposed in utero to sympathetic stress did not resist the administration of this dose of ISO and died by the fourth day of the daily dose of ISO, but none of the control rats treated with ISO did. To verify whether the effect was a consequence of the response to ISO, we implanted a microchip with a pressure transducer in the descending aorta to record the in vivo response of the cardiovascular system to ISO administration. When ISO was administered, there was a rapid increase in heart frequency in both control and stressed animals (Figure 7A). There was also compensation in the pressure because there were no changes in systolic and diastolic pressure. However, in the rats exposed to gestational stress, irregularities in the pressure registration appeared. At a higher magnification of the pressure curve for analysis, the appearance of extrasystoles is evident (shown in a green circle in Figure 7B). Extrasystoles appeared up to the moment that the capacity to regulate pressure was lost, and the rat had a block on the capacity for contraction (Figure 7C). This was probably the reason why rats under gestational stress not only modified the morphological and biochemical process of the heart but also the capacity to respond to an adrenergic overload. 

## 4. Discussion

Most modern theories about stress recognize that although stress is not a disease, it may trigger the majority of diseases when allostatic overload is generated. During stress, glucocorticoids and catecholamines play a key role in the regulation of physiological parameters and homeostasis. The cold stress protocol is classified as a sympathetic stress protocol since this system is only activated, producing the consequent release of NE without a change in the concentration of ACTH or A [10]. This result indicates that there is no activation of the HHA axis [13]. During pregnancy, the placenta acts as an interface between the mother and the fetus, so any alteration in this connection can induce long-term changes in the health of the offspring. In this case, we previously found that applying this cold stress protocol in rats increases the plasma concentrations of NE of the stressed mother, which leads to a lower expression of placental NETs [9]. The decrease in the functionality of the transporter leads the fetus (either male or female) to develop in an environment with high concentrations of catecholamines such as NE, which can lead to long-term effects on offspring when adults. The study of representative groups of 20 and 60 days old was chosen because it represents two stages of development of the rat, prepubescent and adult, respectively (albino rats reach sexual maturity by the fourth week of age either in females or males [27]). Unlike adult rats, prepubescent rats still have an immature sympathetic nervous system in peripheral organs, such as the ovary [28] and the vas deferens [29], where the sympathetic nervous system changes throughout postnatal development up until they reach sexual maturity and the nervous system is already functionally activated. Thus, we can study rats at a range of ages to assess the hormonal conditions that could affect the hypothalamus and the innervation of the peripheral organs.

The levels of NE in the hypothalamus were increased on day 20 and even more on day 60. Intracellular NA was found in the paraventricular nucleus (PVN), where the locus coeruleus (LC) stimulated its release; that is, the NE that was released from the LC to the PVN was primarily measured thanks to direct innervation [30]. Because PVN is essentially the origin of sympathetic innervation to the heart [31], we can hypothesize that the overactivation of this system could affect heart function. The PVN is a multifunctional nucleus because it has also been shown that an elevation of NE levels in the PVN from female rats induces an increase in the synthesis and release of a thyrotrophin-releasing hormone (TRH), and increases NE in the ovary as well as other peripheral organs receiving sympathetic nerves [32].

When analyzing the levels of 5-HT, it must be considered that the endogenous synthesis of 5HT is carried out in the serotoninergic terminals of projections of serotoninergic neurons located in the nucleus of Raphe and projected toward the preoptic area (POA) of the hypothalamus [33]. The increase in 5-HT levels at 20 days old in stressed rats and the recovery when adults were similar to the previous increases with unpredictable stress (movement restriction, food deprivation, or forced swimming) applied to pregnant rats. The authors found that, in their offspring (21-day-old females), there was an increase in 5-HT [34]. However, when measuring the 5-HT levels in adulthood (60 days), there was no further increase in unstressed control rats. It is necessary to emphasize that unpredictable stress or anger-induced stress not only activates the sympathetic system but, in the long term, can generate changes such as sympathetic cold stress. Therefore, on day 20, it can be said that the serotoninergic system is mostly stimulated; one of the possibilities is because the CNS is still in the process of maturation, and it could be hypothesized that this excessive stimulation can be reflected in elevated 5-HT levels [35]. If these early changes influence adulthood, they probably occur during the early stages of development because, after this age, there is no differences in serotonergic activity (60 days old).

### Effects of Gestational Stress in the Heart

β_1_- and β_2_-adrenergic receptors are the subtypes predominantly expressed in the hearts of many mammalian species, including humans, and are the principal regulators of cardiovascular function [36]. Under normal physiological conditions, the β1-adrenergic receptor predominant subtype in cardiomyocytes comprises approximately 80% of the total β-adrenergic receptor, whereas the β2-adrenergic receptor comprises approximately 20%. Regarding β-adrenergic receptors, it has been studied in humans that heart failure produces alterations in the population of β-adrenergic receptors in the ventricular myocardium, causing a decrease in the concentration of β1-adrenergic receptors due to a downregulation mechanism, without observing changes in the β2-adrenergic receptor population. The stoichiometry of the two β-adrenergic receptor subtypes changes from 77:23 in healthy hearts to approximately 60:40 under heart failure conditions mainly caused by the selective downregulation of β1-adrenergic receptor expression [37]. Our results showed a decreased relative abundance of β1-adrenergic receptors without changes in β2-adrenergic receptors in male offspring of prenatally stressed rats. Jevjdovic et al. [38] showed that prenatal stress may induce region- and sex-specific β1- and β2-adrenergic receptor expression patterns within the left ventricle. By using an unpredictable stress paradigm during gestation, they found a decrease in the ratio of β1/β2 receptor that was similar to that found in our work, but they did not find differences in the male progeny. This difference could be because our stress paradigm is a sympathetic stress and the one used by Jevjdovic is a mixed stress that also produces changes in corticoids; thus, they could have a mixed effect that can be opposed as we have previously found with restriction and cold stress [39]. In addition [40], the effect of different stressors could differentially affect the expression of β1 or β2 receptor; diet restriction increased β2 receptors in females but not in males; thus, maternal protein restriction programs cardiac sympathetic activity in a sex-specific manner and might explain the increased susceptibility to ischemia-reperfusion injury in males subject to fetal undernutrition. Similarly, Fernandez-Twinn et al. [7] found that low-protein offspring from stressed prenatal rats have reduced β-adrenergic responsiveness and attenuated adrenergic and insulin signaling, suggesting that intrauterine undernutrition also alters the heart failure risk. The multiple effects found by different stressors opens the possibility for multiple studies to understand how the maternal medium affects the progeny. Our studies may suggest that the changes in the β receptor ratio could be manifested in changes at the heart level and also on the cardiovascular function. 

On the other hand, regarding the cardiac consequences of this prenatal stress, in the present work, we found that the hearts of male rat progeny from stressed mothers did not present changes in the weight of the heart (but probably ventricular hypertrophy), although we found clear changes in adrenergic receptor abundance and the response to a challenge with ISO. These results agree with the preliminary data of our laboratory in which we observed that in the sister of the male progeny, female rats presented a more significant and prolonged cardiovascular response to a stress situation than we previously described in adult females [8]. In addition, they would be more susceptible to β-adrenergic stimulation with isoproterenol [41] than males (125 µg/kg instead of 1 mg/kg in males), indicating that the existence of a higher concentration of receptors gives them greater sensitivity to adrenergic stimulation.

The increase in the IC50 of atenolol, zinterol, and propranolol found in rats exposed to sympathetic stress in utero indicates a lower affinity of drugs for β-adrenergic receptors. Although the decrease in the affinity of the ligand for its receptor could be the result of post-transductional changes in the receptors, the displacement results did not show a decrease in the number of receptors. The decrease in affinity could be explained by protein modification mechanisms, occurring by intrauterine stress. Fewer β_1_-adrenergic receptors and low affinity could result in conditions similar to those found in myocardial infarctions in which there is a decrease in the concentration of β_1_-adrenergic receptors without changes in the concentration of β_2_-adrenergic receptors [42], and they could be associated with changes in the pathological responses of the heart to stress.

On functional grounds, the fact that gestational stress decreased the noradrenergic activity in the heart of the male progeny with no changes in the receptor protein means that either higher sympathetic activity or a higher concentration of β_1_ or β_2_ agonist is needed to obtain normal function. On the contrary, we have decreased NE and β_1_-adrenergic receptor abundance, probably as a compensatory mechanism, and under the lower NE activity, the administration of ISO could act on other receptors, especially at the central level, to modify sympathetic discharge from the PVN and produce an imbalance in systolic pressure and frequency (local effect at the pacemaker) [31]. This condition could generate a pressure response in which the heart cannot maintain a high frequency that finally produces extrasystolic activity and thus noncontrolled heart function once the sympathetic nervous system is activated in the face of a stressful situation, as in this cold stress protocol, the release of NE occurs, which acts on β-adrenergic receptors. In this condition, gestational stress programs β-adrenergic receptors during prenatal development, and the effects of this programming are seen in postnatal development, with the heart being more sensitive to neuronal overload without the possibility to regulate and then weaker to adapt to more stressful conditions. 

Consequently, ISO administration caused death in 50% of rats from stressed mothers during the first 4 days of treatment versus 100% survival of control rats treated with ISO, which prematurely demonstrated that the effect of maternal cold-stress-induced physiological changes in the heart, so that they were more susceptible to adrenergic overload. This greater reactivity to treatment with ISO in the rats stressed in utero could be associated with the change in the affinity of β_1_-adrenergic receptors. In this sense, it was observed that it was associated with a larger size of the left ventricle and a larger area of cardiomyocytes, as shown in our work.

During the stress reaction, the sympathetic adrenal medullary system, and the hypothalamus–pituitary–adrenal cortex axis are activated, causing β-adrenergic receptor overstimulation and remodeling of the β_1_/β_2_-adrenergic receptor ratio in the membrane fraction of cardiac tissue. Cardiovascular disorders have also been related to altered β-adrenergic receptor signaling at or beyond the receptor level. A common feature of many of these conditions seems to be decreased β_1_-adrenergic receptor signaling coinciding with increased β_2_-adrenergic receptor signaling. These may play an adaptive role in the increased sympathetic drive to the heart, protecting the cardiac tissue from the cardiotoxic effects mediated by β_1_-adrenergic receptors without altering cardiac output since this would be sustained by the β_2_-adrenergic receptors that would also have cardioprotective effects against myocyte apoptosis and remodeling. However, the selective maintenance of the β_2_-adrenergic receptor population might help diminish the risk of catecholaminergic overstimulation of the heart since this adrenoceptor subtype couples to Gs and Gi and thus constitutes a part of the “suppressive” actions of glucocorticoids, which limit the stress reaction. 

Moreover, the apoptotic and anti-apoptotic effects of catecholamines mediated by β_1_- and β_2_-adrenergic receptors, respectively, represent the “protective” role of glucocorticoids during the stress reaction [43]. Alterations in the density of cell surface receptors or the affinity of these receptors to different agonists, whether they are endogenous, as well as in the coupling of these receptors to intracellular effectors, are of crucial clinical importance since these processes may alter the functioning of the cardiac tissue and its sensitivity to endogenous mediators or therapeutic drugs. In the human heart atria, increased coupling of the β_1_- and β_2_-adrenergic receptors to their intracellular effectors trigger the processes of arrhythmias. In this regard, our results showed a strong presence of arrhythmias, which could be associated with maladaptive adrenergic stimulation.

Finally, the cardiotoxic effects of catecholamines are mediated via prolonged adrenergic receptor stimulation. Thus, a decrease in β_1_-adrenergic receptors could be associated with a mechanism to protect cardiac function from excessive adrenergic stimulation. Chronic β_1_-adrenergic receptor hyperstimulation by ISO results in their downregulation, and consequently, there is a marked reduction in the inotropic cardiac reserve, as evidenced by frequent tachyarrhythmia leading to death.

## 5. Conclusions

The present study demonstrated that the exposure of the rat’s progeny to catecholamine during fetal development permanently modifies the cardiovascular function, making the rat more susceptible to changes in the catecholaminergic environment and thus decreasing the capacity to respond to changes in the external medium when adults. The identification of the mechanisms that induce such alterations in the β-AR, as well as the protective mechanisms against those alterations, may be of major importance in the clarification of the pathological processes related to stress due to noradrenergic exposure, as well as its therapy, to prevent cardiovascular risk derived from the exposure to stress during gestation.

## Figures and Tables

**Figure 1 ijerph-20-04285-f001:**
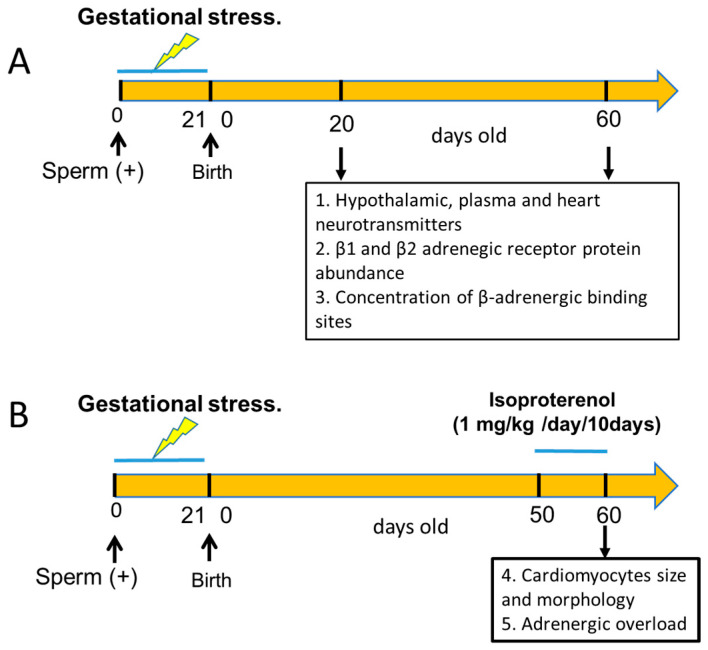
Schematic representation of the experimental groups used in the study. Group A was designed to study the effect of sympathetic gestational stress on the sympathetic neurotransmitter changes in the male progeny and its relationship with β-adrenergic receptor protein expression and the affinity of the β receptors to β ligands. In group B, we tested the effect of gestational stress on the development of cardiomyocytes in the adult heart and the response to an adrenergic overload with a β-agonist in vivo. In (**A**,**B**), we included an equivalent group of control rats that, when pregnant, were not exposed to gestational stress, and the progeny was used as a control.

**Figure 2 ijerph-20-04285-f002:**
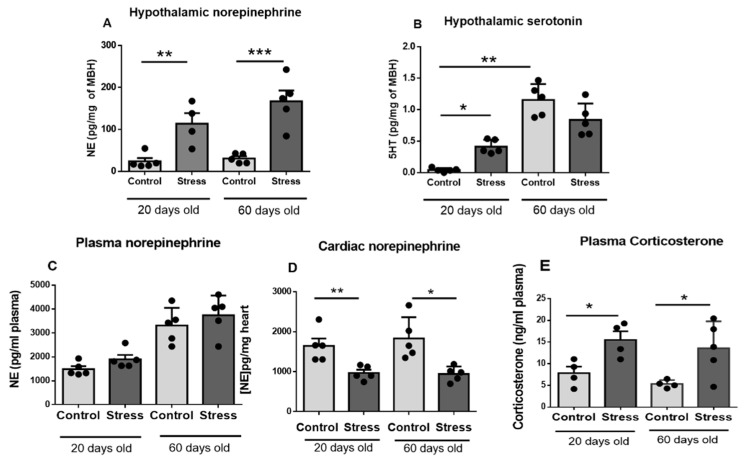
Hypothalamic neurotransmitters norepinephrine (NE) and serotonin (5HT) ((**A**,**B**), respectively), plasma levels of NE (**C**), heart concentration of NE (**D**), and plasma corticosterone (**E**) in male control and stressed pups at 20 and 60 days old. The results correspond to the mean ± SEM of 5 rats in each condition. * *p* < 0.05, ** *p* < 0.1 and *** *p* < 0.001 stress vs. control.

**Figure 3 ijerph-20-04285-f003:**
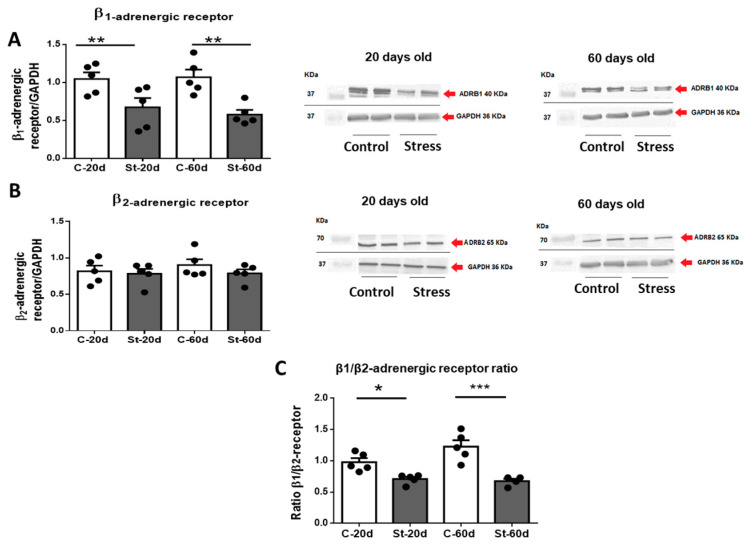
Western blot analysis of β1 adrenergic receptors (Figure 3). On the left side of 3(**A**), the quantification of the signal normalized by the GAPDH signal for 5 control and 5 stressed rats at 20 and 60 days old is shown. ADBR1 represents β1 adrenergic receptor. The results are the mean ± SEM of five rats, each one in control and stressed rats. *** *p* < 0.001 stress vs. control. Figure 3B (left side) shows the mean ± SEM of the data for β2 adrenergic receptor normalized by GAPDH for 5 rats in the control group and 5 in stressed rats at 20 and 60 days old. ADBR2 represents β2 adrenergic receptor. A representative blot is shown on the right side of 3(**A**,**B**), for all the conditions. In the lower panel (**C**), the β1/β2 ratio for the samples above is shown. The results are mean ± SEM of 5 rats in each condition. * *p* < 0.05 stress vs. control, ** *p* < 0.01 stress vs. control *** *p* < 0.001 stress vs. control.

**Figure 4 ijerph-20-04285-f004:**
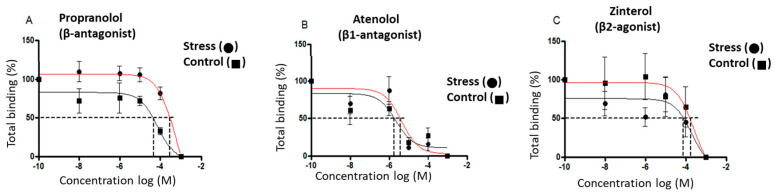
Effect of gestational stress on the displacement of ^3^H-DHA and in the affinity of β receptors from a membrane fraction of heart homogenate from male progeny at 60 days old. In (**A**–**C**), the displacement of the binding of dihydroalprenolol by increasing doses of propranolol, atenolol, and zinterol, respectively, is shown. (**D**,**E**) correspond to the affinity of each of the β ligands and the number of binding sites for each drug, respectively. The results correspond to the mean ± SEM of 6 individual experiments for control and stressed rats. * *p* < 0.05 stress vs. control.

**Figure 5 ijerph-20-04285-f005:**
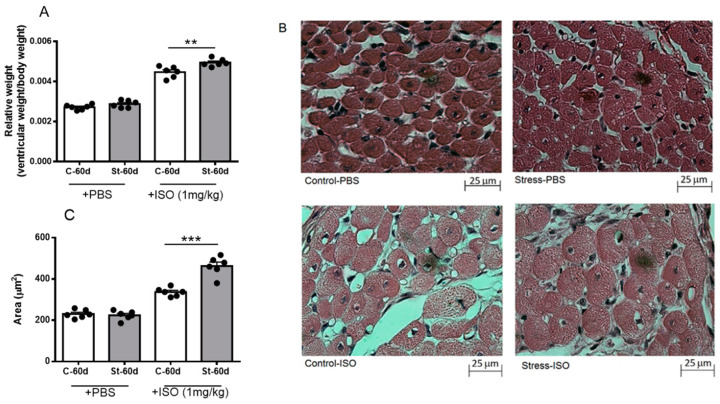
Relative weight (**A**), morphology (**B**), and area (**C**) of cardiomyocytes of 60-day-old male rats after 10 days of isoproterenol or PBS treatment. Hearts were collected at 60 days old after 10 days of treatment with a daily dose of 1 mg/kg of isoproterenol. The results are the mean value ± SEM of 6 animals in each group. The bar indicates the magnification of the picture. A: ** *p* < 0.01 control-ISO vs. stress-ISO; C: *** *p* < 0.001 control-ISO vs. stress-ISO.

**Figure 6 ijerph-20-04285-f006:**
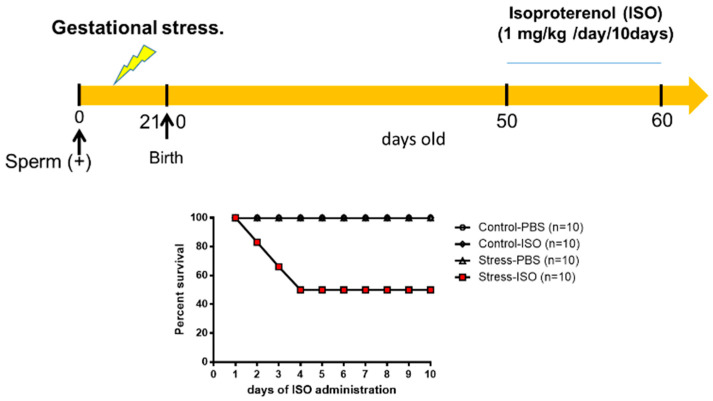
Gestational stress increased the cardiac sensitivity of the male progeny to adrenergic overload. Rats were treated with daily doses of 1 mg/kg weight isoproterenol or saline for control or stressed rats. The graph reflects the percentage of survival of young adult rats after their corresponding treatments. The results correspond to the mean ± mean standard error. Each group consists of 10 animals.

**Figure 7 ijerph-20-04285-f007:**
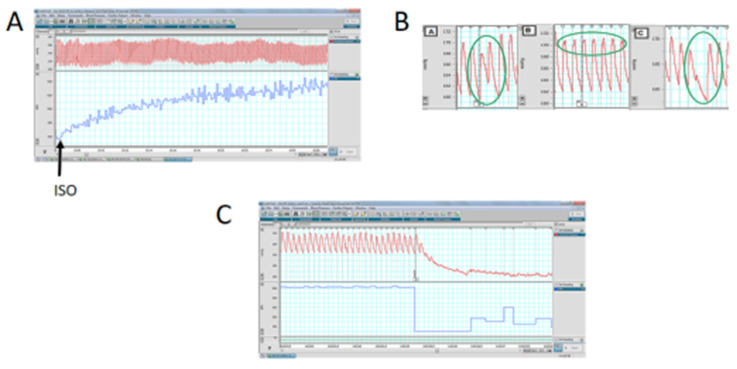
The effect of isoproterenol on the in vivo response in systolic and diastolic pressure (upper part of the graph) and frequency (lower part of the graph). (**A**): The rapid increase in the frequency after administering isoproterenol, without modification of the systolic/diastolic pressure. (**B**,**C**): The response to isoproterenol in one stressed rat, to demonstrate the presence of extrasystoles (highlighted in the green circle in (**B**)) and finally in the failure to recover the heart function in the stressed rats (letter C).

**Table 1 ijerph-20-04285-t001:** General characteristics of the progeny obtained after gestational stress. The number of male rat progeny used for the present work either at 20 or 60 days old according to Figure 1 is shown. The number of 9 or 10 rats at 20 days old considers two series: one was the control with 4 rats, and the other was the progeny of 5 stressed rats. The increased number at 60 days old was due to the groups of control treated with ISO or with saline. Results represent mean values ± SEM with the total n for each group in parenthesis.

	Control	Gestational Stress
Days of gestation	21.8 ± 0.2 (n = 8)	21.9 ± 0.1 (n = 12)
Pups per mother	14.3 ± 0.3 (n = 8 mothers, 100 pups)	12.1 ± 1.1 (n = 12 mothers, 133 pups)
Number of pups by sex		
Females	7.0 ± 0.7 (n = 49 pups)	6.2 ± 0.7 (n = 68 pups)
Males	7.2 ± 0.8 (n = 51 pups)	5.9 ± 0.8 (n = 65 pups)
Males rats used in this study		
Weight of males		
20 days old	53.7 ± 3.0 (n = 9)	54.8 ± 1.4 (n = 10)
60 days old	361.3 ± 11.1 (n = 34)	373.4 ± 12.4 (n = 36)
Heart weight/rat weight		
20 days old	0.005 ± 0.0002 (n = 9)	0.005 ± 0.0001 (n = 10)
60 days old	0.003 ± 0.0001 (n = 14)	0.003 ± 0.0001 (n = 16)

## Data Availability

All data are available under written request.

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
