# Peer review of "Exposure of the Gestating Mother to Sympathetic Stress Modifies the Cardiovascular Function of the Progeny in Male Rats"

_ijerph, 2023, doi:10.3390/ijerph20054285_

Round 1
Reviewer 1 Report
Introduction
1.Lines 52-54 on page 2, after introducing the effect of stress (especially chronic stress) on fetal programming, it is simply mentioned that "as would be in the case of a cold situation without hypothermia", As for why the cold stimulus was chosen for research in chronic stress, the introduction to the issue was not clear enough. It is suggested that the author should make the background more clear.
2.Lines 63-64 on page 2, "This stress modifies the reproductive function of the female progeny" is mentioned in the article. This article is to explore the effects of cold stimulation on the cardiovascular function of male progeny. The literature quoted here seems to be unrelated to the article.
3.Lines 63-70 on page 2, the author cite many animal experiments to introduce the effects of cold stimulation on offspring development, so can some evidence be found from human studies?
Animals and experimental design
1.Lines 100-106 on page 3, the author presented the experimental grouping design scheme, but there is a small question here: how to select the time node of 50 days to inject Isoproterenol? Is there any basis for the injection dose?
2.Lines 225-232 on page 5, the author introduced the measurement methods of area and perimeter of myocardial cells in detail, and the relative weight of myocardial cells was mentioned in the results. It is suggested that how to measure it be described in the Methods section.
3.Lines 241-242 on page 6. According to the equation provided by the authors, the minimum number of animals used in the study is 4.5. Therefore, in fact, each group needs at least 5 animals instead of 4.
Results
1. Figure 2 and Figure 3 on page 7 and 8, the zoom format of the picture does not seem to be adjusted to scale, the font format is inconsistent, and it is recommended to modify the adjustment.
2. Figure 7 on page 11, the "Figure 8" logo in the upper right of Figure 7B seems to be irrelevant to this figure, and it is suggested to be removed.
3.Lines 265-266 on page 7, the authors describe that "No changes due to age were found in control rats." Are there any changes due to age found in stressed rats?
Discussion
1.Lines 455 on page 13, the font format and context are quite different, and there are many font inconsistencies elsewhere in the article. It is recommended to check the font format in the full text.
2.As for the significance and prospect of this study, the author has discussed it in a shallow way, so it is suggested to improve and enrich this part.
Conclusion
1.Lines 524-525 on page 14, is the effect of hypothermia stress on cardiovascular risk broadly representative of most stresses? This sentence in the conclusion seems out of place.
Author Response
The response to referee 1 is in the selected file. Unfortunately it has another name that is different as the original file

Reviewer 2 Report
The authors should work on the suggestions provided by the reviewer.

Author Response
The response to the referee is in the attached file. It does not have the original name but its contains the responses to referee 2

Reviewer 3 Report
I thank the Editor for the opportunity to review this interesting manuscript and the authors for their efforts. The manuscript is well written and the results contribute to this developing field; I suggest only minor revisions, which are listed below:
- Please check your manuscript for typos and unify the font.
- I suggest elaborating on the possible implications of your study for humans in the introduction
- The conclusion should be more in-depth and better structured. I suggest briefly summarizing the main findings of your study and their implications for humans
- There is no indication of the limitations of the study; future directions should also be explored further.
Author Response
The attached file has the responses to referee 3. It does not have the original name but it contains the responses to the referee 3

Round 2
Reviewer 1 Report
none